# Multistable and dynamic CRISPRi-based synthetic circuits

Javier Santos-Moreno [1], Eve Tasiudi [2], Joerg Stelling [2] & Yolanda Schaerli [1]✉

Gene expression control based on CRISPRi (clustered regularly interspaced short palindromic repeats interference) has emerged as a powerful tool for creating synthetic gene circuits, both in prokaryotes and in eukaryotes; yet, its lack of cooperativity has been pointed out as a potential obstacle for dynamic or multistable synthetic circuit construction. Here we use CRISPRi to build a synthetic oscillator ("CRISPRlator"), bistable network (toggle switch) and stripe pattern-forming incoherent feed-forward loop (IFFL). Our circuit designs, conceived to feature high predictability and orthogonality, as well as low metabolic burden and context-dependency, allow us to achieve robust circuit behaviors in *Escherichia coli* populations. Mathematical modeling suggests that unspecific binding in CRISPRi is essential to establish multistability. Our work demonstrates the wide applicability of CRISPRi in synthetic circuits and paves the way for future efforts towards engineering more complex synthetic networks, boosted by the advantages of CRISPR technology.

[1] Department of Fundamental Microbiology, University of Lausanne, Biophore Building, 1015 Lausanne, Switzerland. [2] Department of Biosystems Science and Engineering, ETH Zurich and SIB Swiss Institute of Bioinformatics, Basel, Switzerland. ✉email: yolanda.schaerli@unil.ch

Synthetic biology aims to build artificial decision-making circuits that are programmable, predictable and perform a specific function[1]. Since the rise of synthetic biology in the 2000s, most synthetic circuits have been governed by protein-based regulators. Recently, however, there has been growing interest in circuits based on RNA regulators as a means to overcome some of the intrinsic limitations of protein regulators[2].

The prokaryotic adaptive immunity system CRISPR constitutes a powerful platform for the construction of RNA-driven synthetic circuits[3]. The catalytically-dead mutant dCas9 can be easily directed to virtually any sequence by a single-guide RNA molecule (sgRNA). When a prokaryotic promoter (or downstream) region is targeted, steric hindrance by the dCas9-sgRNA complex results in transcriptional repression—an approach known as CRISPR interference (CRISPRi). CRISPRi offers several advantages over protein regulators for synthetic circuit design. Due to its RNA-guided nature, CRISPRi is highly programmable[4], allows for easy design of sgRNAs that can be highly orthogonal[5] and whose behavior in different environments can be easily predicted in silico[6,7]. It also imposes low burden on host cells[2] and is encoded in shorter sequences than protein-based repressors, thereby facilitating circuit handling and delivery and reducing cost. A potential drawback of CRISPRi is the lack of cooperativity[8]. Cooperative protein transcription factors typically function non-linearly, a difference that might prevent the successful implementation of CRISPRi-based dynamic and multistable circuits[8–10]. Besides, unlike other RNA circuits that exhibit fast dynamics, low rates of dCas9:DNA dissociation in the absence of DNA replication[11,12] effectively slow down CRISPRi dynamics. Potential strategies to overcome such limitations and yield multistable and dynamic circuits have been proposed, and include the precise tuning of active dCas9 degradation, an increased CRISPRi fold-repression, faster dCas9:DNA binding[9], fusing dCas9 to additional repressor domains[13] or incorporating activators in the circuit design[10,14]. Alternatively, competition for dCas9[15] and cellular resources may render non-cooperative systems non-linear, as demonstrated for a T7 RNA polymerase-controlled positive feedback circuit[16].

The last few years have seen a growing interest in developing CRISPRi-based synthetic circuits[8,13,17–25]. However, despite the enormous potential of CRISPRi for synthetic circuit design, the use of CRISPRi circuits in prokaryotes has been largely focused on logic gates and to the best of our knowledge none of the flagship circuits in synthetic biology (namely, the bistable toggle switch[26] and the repressilator[27]) have been re-constructed using CRISPRi. Here, we fill this unaddressed gap by demonstrating that CRISPRi can be used for building some of the most notorious (synthetic) circuit topologies: the repressilator, a bistable toggle switch, and an incoherent feed-forward loop (IFFL, a.k.a. band-pass filter) that drives stripe pattern formation. Our circuit designs, conceived to feature high predictability and orthogonality, as well as low metabolic burden and context-dependency, allow us to achieve robust circuit behaviors in *Escherichia coli* populations. Mathematical modeling suggests that unspecific binding in CRISPRi is essential to establish multistability.

## Results and discussion

**Design features and CRISPRi testing.** To maximize circuit predictability and orthogonality and minimize metabolic burden on the host, we employed a series of design features in all our synthetic networks. The main circuit components were all expressed from a single vector (so-called variable vector) to avoid fluctuations in their stoichiometry. Circuit transcriptional units (TUs) were isolated from each other by strong transcriptional terminators and 200 bp spacer sequences, to avoid transcriptional

readthrough and minimize compositional context[28]. To prevent mRNA context-dependency and provide transcriptional insulation within TUs, a 20 bp csy4 cleavage sequence[29] was used flanking sgRNAs and upstream of the ribosome binding sites (RBS) of the reporter genes. Csy4 is an RNase of the CRISPR system of *Pseudomonas aeruginosa* that processes the transcript of the CRISPR array to release the CRISPR-derived RNAs (crRNAs)[30]. Here, we employed Csy4 processing to release parts that are transcribed together in the same mRNA molecule but need to act independently once transcribed—e.g., sgRNAs that need to bind dCas9 for function, and fluorescent reporter transcripts that have to be translated. To avoid cross-talk between fluorescent reporters, orthogonal degradation tags[31] were employed. Sequence repetition at the DNA level was minimized to prevent unwanted recombination events. The levels of dCas9 and Csy4 were kept constant by expressing them from constitutive promoters in a separate constant vector. All gene circuits were tested in *E. coli* (MK01[32]) incubated in a rich defined medium (EZ, Teknova) for maximizing cell fitness while reducing variability. In order to speed up the design-build-test cycle, we adopted a previously described cloning strategy that allows for fast and modular assembly of synthetic networks[33].

We first devised a simple circuit to assess the dose-response of CRISPRi repression. This 2-node network design displays Boolean NOT logic, in which node 1 (N1) is induced by L-arabinose (Ara) and represses expression of node 2 (N2) (Fig. 1). Levels of N1 and N2 in the NOT circuit were monitored via the fluorescent reporters mKO2 and sfGFP (Fig. 1b) carrying degradation tags (MarA and MarAn20) to accelerate reporter turnover[31]. We chose four single-guide RNAs (sgRNA-1 to -4) and their corresponding binding sites (bs-1 to -4), previously shown to be orthogonal to each other[17]. All binding sites were placed at the same distance from the promoter to prevent differences in the repression derived from binding site position[5]. Maximal induction (0.2% Ara) of N1 produced sgRNA levels that repressed the expression of N2 in the range of 10 to 30-fold (Fig. 1c), with sgRNA-2 being the strongest repressor. Truncation of the four 5′ nucleotides ("t4") of sgRNA-1 and -4 reduced maximal repression by 20 and 45%, respectively, providing a means to tune down repression (Fig. 1c). We also measured N2 fluorescence in the presence of different Ara concentrations (Fig. 1d), which showed a modulable, dose-dependent CRISPRi repression, a prerequisite for building dose-dependent circuits.

**A CRISPRi toggle switch.** We next sought to build a classical genetic circuit that has never been constructed before using CRISPRi: the toggle switch (TS)[26], where two nodes mutually repress each other to produce bistability, i.e., two mutually-exclusive stable states. We designed a TS circuit in which two nodes produce sgRNAs that repress each other (Fig. 2), resulting in two non-concurrent stable states: HIGH, in which the fluorescent reporter sfGFP-MarAn20 is produced at high levels, and LOW, in which sfGFP-MarAn20 is expressed at negligible levels. The two states can be toggled through a controller plasmid, in which the addition of Ara or AHL (N-(β-Ketocaproyl)-L-homoserine lactone) triggers the expression of sgRNA-1 or -4, thus favoring the LOW and the HIGH states, respectively (Fig. 2b). The TS was assessed by incubating cells in four successive media containing the first inducer (Ara or AHL), followed by no inducer, then the second inducer (AHL or Ara), and finally no inducer again (Fig. 2c, d). Most cells adopted the LOW state following an initial incubation with Ara and kept this state even when Ara was removed. When AHL was added, cells switched to the HIGH state and kept this state after AHL removal (Fig. 2c left, and d). The reverse induction scheme (AHL-nothing-Ara-nothing) also

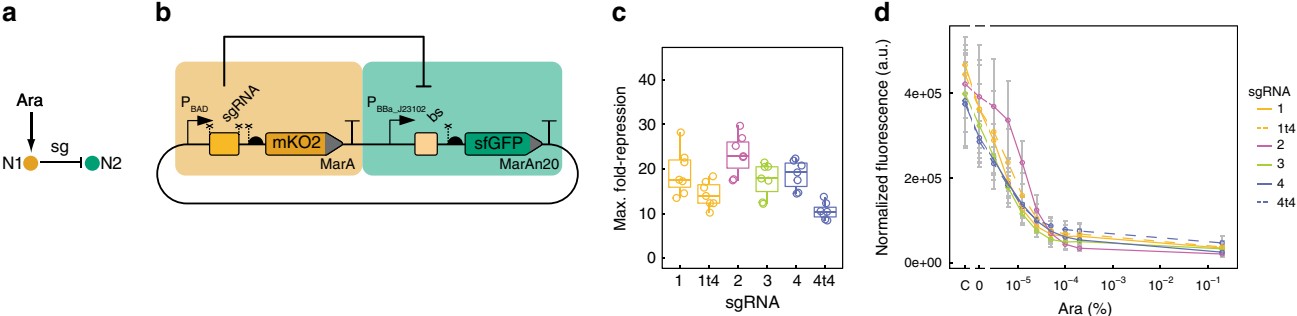

**Fig. 1 Assessment of CRISPRi repression through a 2-node NOT logic circuit. a** Schematic representation of the circuit architecture, where the first node (N1) is induced by Ara and in turn produces a sgRNA (sg) that represses the second node (N2). **b** Details of the circuit design. In addition, AraC, dCas9, and Csy4 are constitutively expressed. Bent arrows: promoters; squares: sgRNA binding sites; rectangles: sgRNAs; crosses: csy4 recognition sites; semicircles: RBSs; pointed rectangles: reporter genes with degradation tags; and T-s: transcriptional terminators. **c** Maximum fold-repression of N2 achieved by four different sgRNAs, calculated as the GFP fluorescence of a fully-induced (0.2% Ara) circuit relative to a control lacking the sgRNA. Truncated variants of sgRNA-1 and -4 lacking the four 5′ nucleotides (t4) were also assessed. Data correspond to seven biological replicates. Boxes represent quartiles Q1 (box lower limit), Q2 (i.e., median, internal line) and Q3 (box upper limit); whiskers comprise data points that are no more than 1.5× IQR (inter-quartile region, i.e., the length of the box) away from the box. **d** Dose-response curves for the NOT circuits operating through sgRNA-1 to -4 (including t4 variants of sgRNA-1 and -4). The response (GFP fluorescence) of all four circuits can be modulated by adjusting the input dose (Ara). The controls lacking the sgRNAs (C) showcase some level of P$_{BAD}$ leakiness in the NOT circuits that results in weak repression even in the absence of Ara (0). Mean and s.d. of three biological replicates. Source data are provided as a Source Data file.

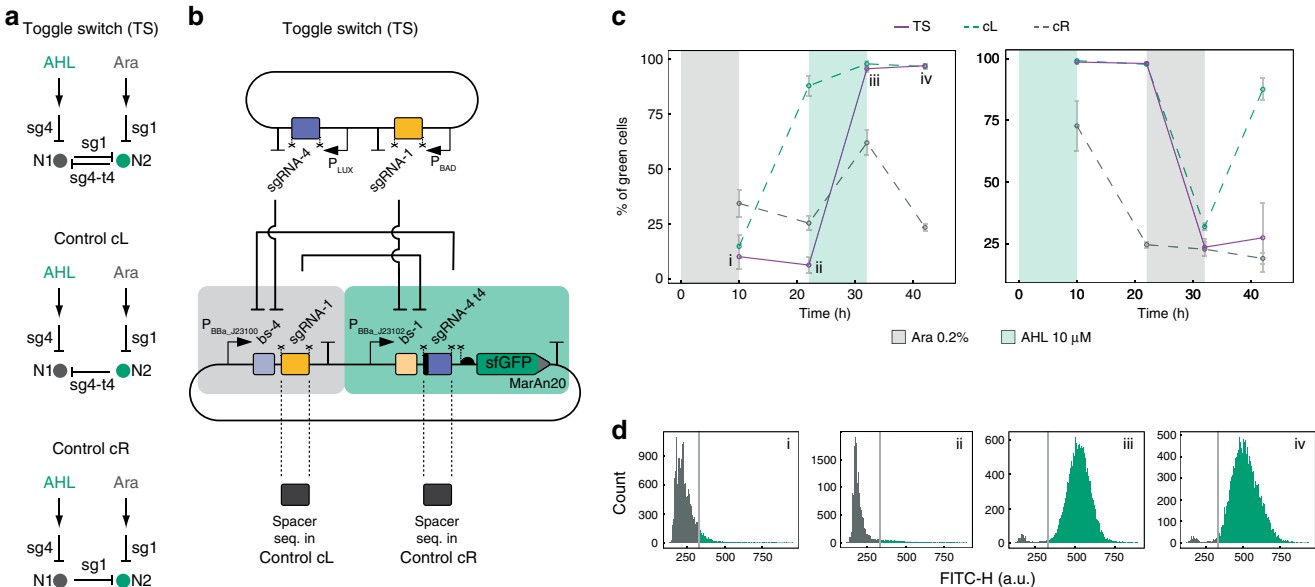

**Fig. 2 The CRISPRi-based toggle switch (TS). a** Topology of the bistable TS and the monostable controls. **b** Molecular implementation. Symbols as described in Fig. 1. **c** Behavior of TS and control circuits over time in response to two induction schemes: Ara-nothing-AHL-nothing (left) or AHL-nothing-Ara-nothing (right). The percentage of green cells was determined by flow cytometry for four biological replicates (shown are the mean and the s.d.). **d** Histograms showing representative distributions of the TS cells according to their green fluorescence for the four timepoints indicated in panel **c**: (i) after Ara induction, (ii) after Ara followed by no inducer, (iii) after Ara-nothing-AHL, and (iv) after Ara-nothing-AHL-nothing. Black: cells below threshold for 'HIGH' state (gray line); green: cells in 'HIGH' state. Source data are provided as a Source Data file.

displayed bistability and hysteresis (Fig. 2c). Conversely, the control circuits lacking one of the repressors (Fig. 2a) failed to display any bistability: both cL (biased towards the HIGH state) and cR (biased towards the LOW state) showed a monostable behavior and were unable to keep the LOW and the HIGH states in the absence of inducer, respectively (Fig. 2c). Together, these results show that CRISPRi can be applied to build multistable gene circuits.

**Origin of bistability.** Because CRISPRi's non-cooperative binding[8] contrasts with the requirement for cooperativity in the protein-based TS[26], we aimed to explain how CRISPRi can generate multistability. Theory shows that cooperativity is sufficient, but not necessary for a two-gene circuit with mutual inhibition to be bistable; alternative mechanisms include a significant depletion of free repressor[34]. We considered this a plausible mechanism for CRISPRi because of dCas9's scanning of PAM sites—the effective affinity of binding to sites not matching the sgRNA sequence is comparatively low[35] (considered here as unspecific binding), but PAM sites are abundant in the genome. We therefore developed mechanistic mathematical models in the form of reaction networks with mass-action kinetics that comprise dCas9/sgRNA binding to DNA, as well as production and degradation of all components (see Supplementary Figs. 7–12 for

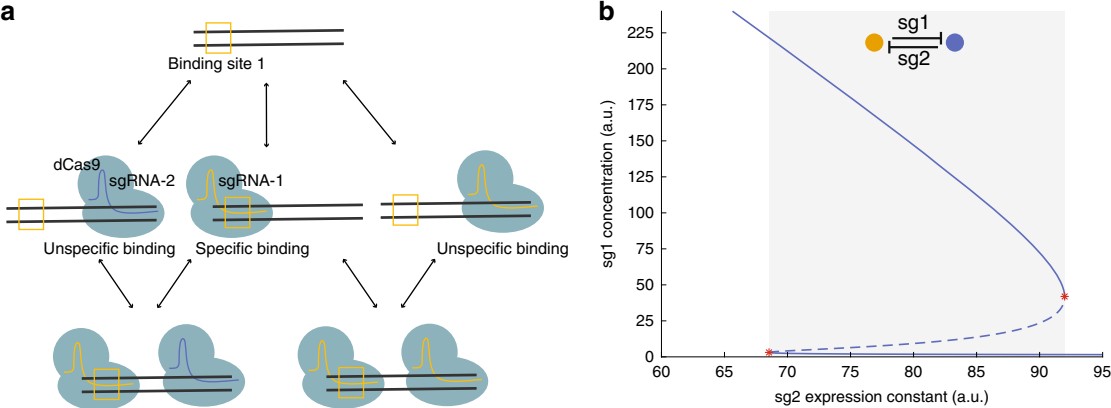

**Fig. 3 Proposed origin of bistability in the CRISPRi TS. a** Binding configurations of two dCas9/sgRNA variants (colors). We assume a specific binding site for one variant (box) and unspecific binding during screening of PAM sequences for both variants. Arrows: binding/unbinding reactions. **b** Bifurcation diagram for the mathematical model with specific and unspecific dCas9/sgRNA binding (biologically constrained parameters; see Supplementary Fig. 12 and Supplementary Table 4). Solid (dashed) lines: stable (unstable) steady states; red stars: tangent bifurcations; gray: bistability region. Note that sgRNA-1 (abbreviated sg1) and sgRNA-2 (sg2) in the model represent two generic sgRNAs, and not necessarily the specific sequences labeled as sgRNA-1 and sgRNA-2 in the experimental parts of this work.

details). We then used methods from chemical reaction network theory[36,37] to evaluate the potential of a network structure for bistability, without prior knowledge on parameter values. Consistent with theory, we did not find bistability for networks with specific binding only. However, network extensions by unspecific binding to target genes, with independent (that is, non-cooperative) specific and unspecific binding events (Fig. 3a) could show bistability. This held even when unspecific binding did not inhibit gene expression and when we constrained parameters to experimentally determined ranges (Supplementary Table 4), as illustrated in Fig. 3b. Thus, our model-based analysis suggests that the interplay between specific and unspecific binding of dCas9/ sgRNA complexes to DNA explains bistability in the CRISPRi TS.

**CRISPRi incoherent feed-forward loops**. We next decided to build another flagship synthetic circuit, a stripe-forming IFFL (Fig. 4)[38–40]. We designed a CRISPRi-based 3-node IFFL type 2 (I2), which relies exclusively on repression interactions[41]. To monitor the behavior of all three nodes of the circuit, we employed compatible fluorescent reporters fused to orthogonal degradation tags[31]: mKO2-MarA, sfGFP-MarAn20 and RepA70-mKate2 (Fig. 4b). The rationally designed circuit performed as expected: with increasing Ara levels N1 levels raised, resulting in a concomitant decrease of N2; a peak (stripe) of N3 was observed in the open window between N1 and N2 (Fig. 4c). Our three-reporter IFFL allowed us to readily monitor network behavior and to optimize it—and to use the circuit as a concentration-detector (Supplementary Fig. 1). To our knowledge, no IFFL with three reporters had been constructed before. The inclusion of three reporter protein-coding genes was enabled by the small size and transcriptional nature of the circuit, causing low burden on host cells. Even more, due to the relatively small size of the circuit in terms of DNA sequence, all 3 nodes of the network could be carried within a single plasmid, thus keeping their relative abundance constant and avoiding fluctuations in their stoichiometry due to plasmid copy number variations. To corroborate that the CRISPRi IFFL was capable of true spatial patterning, we homogeneously spread bacteria on an agar plate and applied a central source of arabinose, which diffused forming a radial gradient. Consistent with the French flag model[42], the initially identical, isogenic bacterial population carrying the synthetic network interpreted the Ara gradient into three discrete gene expression programs: orange, green and blue (Fig. 4d).

We next wanted to explore the robustness of the CRISPRi stripe circuit with respect to variations in its molecular components. We assessed the performance of (i) a circuit carrying a single reporter (sfGFP-MarAn20 in N3), (ii) a circuit with a different set of sgRNA regulators (sgRNA-4, sgRNA-4t4 and sgRNA-3 instead of sgRNA-1, sgRNA-1t4 and sgRNA-2), and (iii) a network with swapped N2 and N3 reporters (Supplementary Fig. 2). All these circuits showed a stripe behavior (Supplementary Fig. 2), demonstrating the robustness of our CRISPRi-based IFFL, which is enabled by our design aiming at high orthogonality and low context-dependency.

An important limitation that commonly affects synthetic gene circuits is the difficulty to expand their functionality by including existing circuits into more complex networks or by making them operate in a different environment, such as in co-existence with another circuit. To test whether a CRISPRi-based IFFL could operate in a similar manner in a different context, we decided to build two orthogonal CRISPRi IFFLs that would operate in parallel within the same bacterial cell, each producing a single (and independent) stripe in response to a common arabinose gradient (Fig. 4e). When bacteria containing both parallel circuits were submitted to different Ara concentrations, a double peak of gene expression could be observed at intermediate inducer levels corresponding to two overlapping peaks, each generated by one of the circuits, as designed (Fig. 4f). Importantly, the doubling of the CRISPRi circuit (i.e., two stripe networks instead of one) did not result in a growth defect compared to bacteria carrying only one of the IFFLs or a control without IFFL circuits (Supplementary Fig. 3). To our knowledge, no other study has reported to date a double synthetic stripe driven by a single (isogenic) cell population. The simultaneous yet independent operation of the double-stripe system was enabled by the intrinsic properties of CRISPRi (orthogonality, low burden, short encoding sequences) and the low context-dependency of the design.

We also assessed the combination of two CRISPRi circuits exhibiting different functions within the same cellular environment. We combined a stripe-forming IFFL with an orthogonal double-inverter circuit[33], a concatenation of two NOT gates that inverts twice the input signal to give an output that mimics the input. This design can be used when the output needs to resemble the input with some additional modification, e.g., a shifted position[33] or a time delay[13]. Bacteria carrying the two circuits perform both functions: an IFFL-generated stripe pattern and a

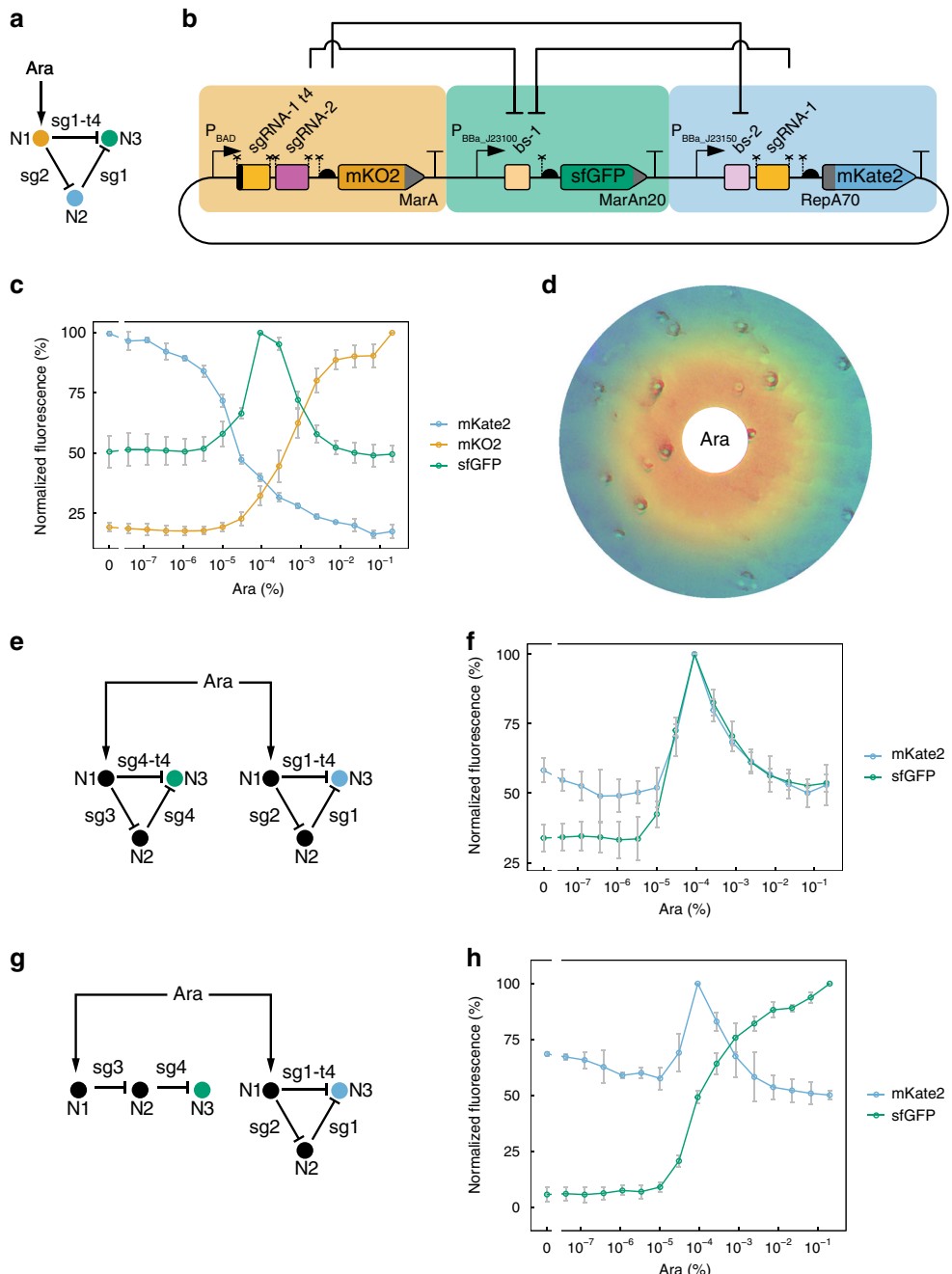

**Fig. 4 Stripe-forming synthetic IFFL based on CRISPRi. a** I2 topology of the circuit design. **b** Molecular implementation of the synthetic CRISPRi IFFL. Symbols as described in Fig. 1. **c** Behavior of the network (output) as a function of inducer concentration (input). Fluorescence of the three nodes was recorded and plotted together. Mean and s.d. of three biological replicates. **d** Spatial pattern formation by the synthetic CRISPRi circuit. Bacteria carrying the synthetic network were plated homogeneously on agar and a paper disk soaked with Ara was placed in the center of the plate (white), creating a radial Ara gradient by diffusion. The initially identical isogenic population developed a 3-domain pattern in which each domain is characterized by differentially expressed genes. Orange, mKO2; green, sfGFP; blue, mKate2. Note that false colors were chosen for mKO2 (emission λ 565 nm) and mKate2 (emission λ 633 nm) to highlight differences between the two. **e** Topology of the two-IFFL system. **f** Double-stripe phenotype displayed by bacteria carrying the two parallel but independent CRISPRi stripe networks. Mean and s.d. of three biological replicates. **g** Topologies of an orthogonal double-inverter and an IFFL circuit. **h** Bacteria carrying the two CRISPRi circuits shown in **g** display both functions. Mean and s.d. of three biological replicates. Source data are provided as a Source Data file.

double-inverter-driven behavior that mimics the Ara gradient (Fig. 4g, h). Thus, the two networks behaved as expected, demonstrating the possibility to combine multiple CRISPRi circuits with distinct functions in the same bacteria.

We also designed an IFFL in which both the control and the reporting are RNA-based. To this aim, we modified our CRISPRi

IFFL (in which the control is sgRNA-based) to remove the orange and blue reporters and substituted the original green reporter (sfGFP) with 8 copies of the dBroccoli (dimeric Broccoli) RNA aptamer[43] (Supplementary Fig. 4). When we incubated the engineered bacteria with the DFHBI-1T fluorophore we observed a stripe of fluorescence (Supplementary Fig. 4b), showing that the

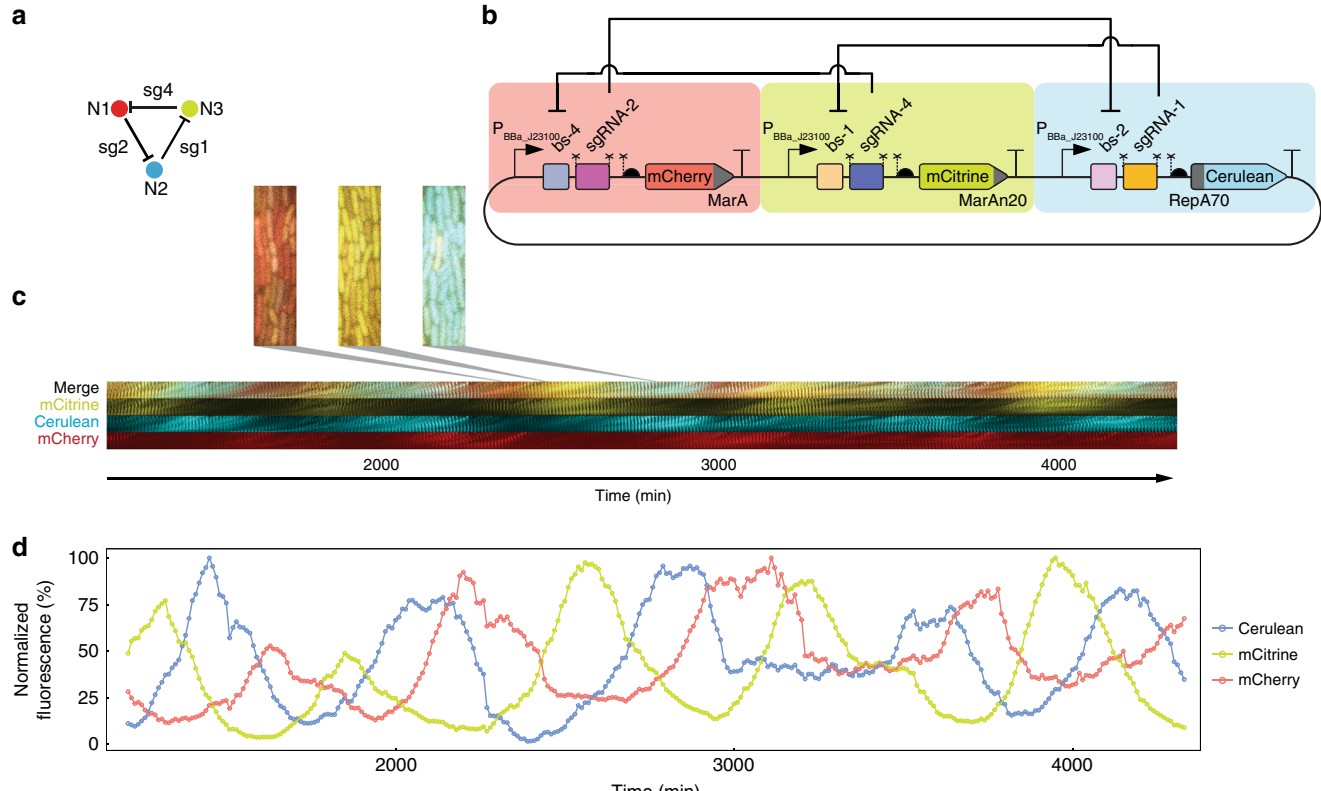

**Fig. 5 The CRISPRlator, a CRISPRi-based oscillator. a** Topology of the ring oscillator. **b** Molecular implementation of the CRISPRlator. Symbols as described in Fig. 1. **c** Montage showing the oscillations of the three fluorescent reporters over time. Bacteria were grown in continuous exponential phase in a microfluidic device for 3 days and imaged every 10 min. Microscopy images (as those enlarged in the zoom-ins) are displayed together in a timeline montage (kymograph). **d** Quantification of the population-level fluorescence of ~110 cells over time. Oscillations display a period of 10–12 h. Source data are provided as a Source Data file.

band-pass behavior of an RNA-controlled IFFL can indeed be reported by an RNA molecule.

**A CRISPRi oscillator (The CRISPRlator).** Finally, we wanted to explore whether CRISPRi-based circuits could be employed not only for spatial patterning, as demonstrated above, but also to create temporal patterns. Oscillations are temporal patterns with key roles in many different biological phenomena. We sought to engineer an oscillatory pattern controlled by a dynamic CRISPRi circuit. To this aim, we designed a 3-node ring oscillator: the CRISPRlator (Fig. 5), in honor of the first synthetic oscillator, called the repressilator[27]. Few (2–3) bacteria carrying the CRISPRlator were loaded per microfluidic chamber (Supplementary Fig. 5), grown there continuously for 3 days (over 100 generations) and imaged every 10 min. The CRISPRlator generated robust oscillations between three states (red, green and blue) with a period of 10–12 h (14–17 generations, Fig. 5c, d), while a control with an open-ring topology did not show any oscillations (Supplementary Fig. 6). After the chambers were filled with cells (~110 cells/chamber), we observed synchronous long-term oscillations, even in the absence of any active cell communication or synchronization mechanism (Supplementary Movie 1). We hypothesize that the observed synchrony stems from the robust inheritance of the oscillatory state across cell divisions from the few cells initially seeded in the microfluidic chamber. This is reminiscent of improved versions of the protein-based repressilator[44,45], where synchronous oscillations were observed due to lower levels of molecular fluctuations and consequently reduced phase drift and increased inheritance of the period compared to the original repressilator[27].

Together, our results demonstrate that the lack of cooperativity of CRISPRi is not an insurmountable obstacle for (complex) synthetic circuit construction and CRISPRi can effectively be used for generating dynamic and multistable behaviors. Specifically, our mathematical analysis suggests that multistability requires unspecific binding (i.e., dCas9's scanning of PAM sites) in CRISPRi. The benefits inherent to CRISPRi circuits will likely prompt the construction of new and extended synthetic systems, including designs hard to achieve with protein repressors, e.g., two independent clocks operating within the same cell. The universality of CRISPR should facilitate the transfer of our circuits into other species. We anticipate that our results will encourage the synthetic biology community to employ CRISPRi for gene circuit design and inspire future construction of more complex synthetic networks.

## Methods

**Circuit construction**. Genes encoding sfGFP, mKO2, mKate2, Csy4, and dCas9 were obtained as previously described[33]. mCherry and mCitrine were amplified from plasmids (pLacI(r)-mCherry(ASV) and pCI-Citrine-(ASV)) kindly provided by Sebastian Maerkl[44]. dBroccoli was amplified from plasmid pET28c-F30-2xdBroccoli, which was a gift from Samie Jaffrey (Addgene plasmid #66843). Cerulean[46] was purchased as an *E. coli* codon-optimized gBlock from IDT, and LuxR (BBa_C0062) was also codon-optimized and synthesized by GenScript. Primers were purchased desalted from Microsynth or Sigma-Aldrich (Supplementary Table 1). The circuits were constructed employing a previously described Gibson-based cloning framework that allows for the fast and modular cloning of synthetic gene networks[33]. Briefly, the method consists of two steps: step 1 involves Gibson assembly of transcriptional units into individual intermediate plasmids; in step 2, these plasmids are digested with restriction enzymes so that the resulting flanking regions contain overlaps that drive a second Gibson assembly into a single plasmid to yield the final circuit. For step 1, all DNA parts carried the same Prefix (CAGCCTGCGGTCCGG) and Suffix (TCGCTGGGACGCCCG) sequences for

modular Gibson assembly using MODAL[47]. Basically, forward and reverse primers annealing to Prefix and Suffix sequences, respectively, were used for PCRs that added unique linkers to the DNA parts. PCR amplifications were column-purified using the Monarch PCR & DNA Cleanup Kit (NEB), and assembled using NEBuilder HiFi DNA Assembly Master Mix (NEB, 1 h 50 °C) into backbones previously digested with corresponding restriction enzymes (NEB, 1 h 37 °C) to yield intermediate plasmids containing individual transcriptional units. In step 2, these intermediate plasmids were digested with enzyme sets yielding overlapping sequences, purified and assembled as described above. One microliter of non-purified Gibson reaction was transformed into 50 μl of electrocompetent NEB5α cells, and 2/5 of them were plated onto selective agar plates. Plasmids used in this study (Supplementary Table 2) are available through Addgene [https://www.addgene.org/Yolanda_Schaerli/] and all sequences are provided in a supplementary file (Supplementary Data 1).

**Microplate reader experiments**. Gene expression of fluorescent reporters was used to assess synthetic circuit performance; fluorescence was measured in microplate readers (except for the flow cytometry in Fig. 2c, d, the agar plate experiment in Fig. 4d and the microfluidic experiment in Fig. 5). MK01[32] electrocompetent cells were transformed with a constant plasmid encoding proteins required for circuit function (namely dCas9 and Csy4, plus LuxR when applicable), as well as with a variable vector bearing AraC (when needed) and the CRISPRi circuit (see Supplementary Table 2 for a detailed plasmid description). Two mili-liter of selective LB were inoculated with single colonies and incubated at 37 °C for ~6 h with 200 rpm shaking; cells were pelleted at 4000 rcf and resuspended in selective EZ medium (Teknova) containing 0.4% glycerol. One hundred and twenty microliter of 0.05 OD$_{600}$ bacterial suspensions were added per well on a 96-well CytoOne plate (Starlab), and 2.4 μl of L-arabinose (Sigma) were added to yield the indicated concentrations. Plates were incubated at 37 °C with double-orbital shaking in a Synergy H1 microplate reader (Biotek) running Gen5 3.04 software. Fluorescence was determined after 16 h (for protein reporters) or 6 h (for Broccoli RNA reporter) with the following settings: mKO2: Ex. 542 nm, Em. 571 nm; sfGPF: Ex. 479 nm, Em. 520 nm; mKate2: Ex. 588 nm, Em. 633 nm. For the Broccoli RNA aptamer, 40 μM DFHB1-1T (Tocris Bioscience) were added to the medium. Fluorescence levels were treated as follows: (i) the fluorescence signal in a blank sample was subtracted, (ii) the resulting value was divided by the absorbance at 600 nm to correct for differences in bacterial concentration, and finally (iii) the bacterial auto-fluorescence of a strain with no reporter genes was subtracted. Subsequently, corrected fluorescence was normalized to a percentage scale by dividing all values of a given color by the highest value of that color. Normalized data were plotted in R[48] using RStudio 1.0.143 (running R 3.4.0).

**Flow cytometry**. MK01[32] electrocompetent cells were transformed with a constant vector encoding dCas9, Csy4 and LuxR plus the controller vector (which carries P$_{BAD}$-controlled and P$_{LUX}$-controlled sgRNA-1 and -4, respectively, plus AraC) together with the TS circuit or the control networks cL or cR. Single colonies were used to inoculate 2 ml of selective EZ medium (Teknova) containing 0.2% Ara or 10 μM AHL and incubated at 37 °C for 10 h in a tilted 200 rpm shaker. Next, cells were pelleted at 5000 rcf for 10 min and resuspended in selective EZ medium free of inducers; 1 μl of the resuspension was used to inoculate 120 μl of inducer-free selective EZ in a 96-well plate, which was incubated at 37 °C for 12 h with double-orbital shaking in a Synergy H1 microplate reader (Biotek). Cells were then diluted 1:120 in 120 μl of selective EZ containing the opposed inducer (10 μM AHL or 0.2% Ara) and grown for 10 h more in the microplate reader under identical conditions. Cells were then diluted into an inducer-free medium as before (resuspension in inducer-free selective EZ followed by 1:120 dilution in the same medium) and incubated for another 12 h in the microplate reader under identical conditions. At the end of each induction period (inducer 1, no inducer, inducer 2, no inducer) a sample was taken, diluted 1:100 in phosphate-buffered saline (PBS) and analyzed with a BD LSRFortessa flow cytometer using a 488 nm laser in combination with FITC filter for sfGFP fluorescence determination. 20,000 events were acquired with BD FACSDiva 8.0 software and data were exported using FlowJo 10.5.2 (FlowJo, LLC). Green cells were gated (green > 330 FITC-H a.u.) in the FITC-H histogram to differentiate the two states (green and non-green) using non-fluorescent cells as control. Both the gating and the plotting were performed in R[48] using RStudio 1.0.143 (R 3.4.0).

**Agar plate assay**. MK01[32] electrocompetent cells were transformed with the constant plasmid pJ1996_v2[33], which carries dCas9 and Csy4, and a variable plasmid encoding the three-color IFFL (pJ2042.2). Two mililiter of selective LB were inoculated with single colonies and grown at 37 °C for ~6 h; cells were pelleted at 4000 rcf and resuspended in selective EZ medium (Teknova) with 0.4% glycerol. Three hundred microliter of 0.15 OD$_{600}$ bacterial suspensions were added to pre-warmed (1 h 37 °C) Petri dishes containing selective EZ medium (Teknova) with 0.4% glycerol and 0.9% agar, and suspensions were spread with sterile glass beads. After incubating 1 h at 37 °C, a filter paper disk was placed in the center of the agar, and 15 μl of 5% Ara were delivered onto the disk. Plates were incubated at 37 °C and imaged after 20 h with an Amersham Typhoon 5 Biomolecular Imager (GE Healthcare) running Amersham Typhoon 1.1.0.7 using the following settings:

mKO2: Ex. 532 nm, Em. 560–580 nm; sfGPF: Ex. 488 nm, Em. 515–535 nm; mKate2: Ex. 635 nm, Em. 655–685 nm. Grayscale images were converted to color images using Fiji[49] 2.0.0 and overlaid.

**Microfluidic experiments**. MK01[32] electrocompetent cells were transformed with the constant plasmid pJ1996_v2[33] and a variable plasmid encoding the CRISPRlator. Single colonies were used to inoculate 5 ml of selective LB, which were grown over-night at 37 °C. Next morning, 3 ml of selective EZ containing 0.85 g l$^{-1}$ Pluronic F-127 (Sigma) were inoculated with 30 μl of the overnight preculture and grown for 3–4 h at 37 °C. Cells were centrifuged for 10 min at 4000 rcf and resuspended in ~10 μl of the supernatant to obtain a dense suspension, which was loaded into the PDMS micro-fluidics device. Cells were grown in a continuous culture inside microfluidic chambers (dimensions: 1.2 μm × 12 μm × 60 μm, h × w × l, purchased from Wunderlichips, see design in Supplementary Fig. 5) for 3 days with a constant 0.5 ml h$^{-1}$ supply of fresh medium (selective EZ plus 0.85 g l$^{-1}$ Pluronic F-127) and removal of waste and excess of bacteria, powered by an AL-300 pump (World Precision Instruments). Imaging was performed using a Leica DMi8 microscope and a Leica DFC9000 GT camera controlled by the Leica Application Suite X 3.4.2.18368, with the following settings: Cerulean: Ex. 440 nm 10% 50 ms, Em. 457–483 nm; mCitrine: Ex. 510 nm 10% 50 ms, Em. 520–550 nm; mCherry: Ex. 550 nm 20% 200 ms, Em. 600–670 nm. Imaging started after 20 h to allow cells to fill the chamber and oscillations to stabilize, and images were collected every 10 min with LAS X software, and analyzed using Fiji[49] for both quantification and montage. Fluorescence quantifications were normalized as follows: background fluorescence was removed by subtracting to each channel the minimum value of that channel, and data were normalized to a percentage scale by dividing all values of a given channel by the highest value of that channel. Normalized data were plotted in R[48] using RStudio 1.0.143 (R 3.4.0).

**Mathematical modeling of the CRISPRi toggle switch**. We performed all ana-lyses in Matlab 2018b (MathWorks, Natick, MA) and used the BioSwitch toolbox[36] version 1.0.0 (available at https://github.com/ireneotero/BioSwitch) to evaluate network structures that could potentially generate bistability. Model structures are defined in Supplementary Tables 3, 4 and Supplementary Figs. 7–13. We per-formed parameter exploration for a limit point in the range $[10^{-2} \ 10^2]$ for all parameters, except for the simplest model (Supplementary Fig. 7; ranges $[10^{-3} \ 10^3]$) and for the most complex model (Supplementary Fig. 12) with experimen-tally constrained parameter ranges (Supplementary Table 4). Code for the analysis is available as Supplementary folder.

**Reporting summary**. Further information on research design is available in the Nature Research Reporting Summary linked to this article.

## Data availability
The source data underlying Figs. 1, 2, 4 and 5 and Supplementary Figs. 1–4 and 6 are provided as a Source Data file. The plasmids used in this study (Supplementary Table 2) and their annotated sequences (Supplementary Data 1) are available through Addgene [https://www.addgene.org/Yolanda_Schaerli/] 124421, 124422, and 140664–140689. All other data are available from the authors upon reasonable request.

## Code availability
The BioSwitch toolbox[36] is available at https://github.com/ireneotero/BioSwitch. The code for the analysis is available as a Supplementary folder within the Source Data file.

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

## Acknowledgements

We thank Imre Banlaki for cloning *luxR* gene into pJ1996_v2 to yield pJ2018, Içvara Barbier for help with microfluidics and feedback about bistable circuits, and Marc García-Garcerà for help with R language. We also thank Mariapia Chindamo, Aysun El Wardani, Florence Gauye, Virginie Kahabdian, Borany Kim and Léo Moser for excellent technical assistance; Marc Güell, Mark Isalan, and Jan-Willem Veening for critical reading of the manuscript, and all Schaerli lab members for useful discussions. This work was funded by Swiss National Science Foundation Grant 31003A_175608.

## Author contributions

J.S.M. and Y.S. designed the experimental research. J.S.M. performed experiments, analyzed data, and prepared the corresponding figures. E.T. and J.S. performed the mathematical modeling, and ET prepared the modeling figures. J.S.M., Y.S., E.T., and J.S. wrote the manuscript. All authors have given approval to the final version of the manuscript.

## Competing interests

The authors declare no competing interests.
