## [Peer Review File · Nature Communications]

Reviewers' Comments:

Reviewer #1:

Remarks to the Author:

Manuscript by Santos-Moreno and Schaerli describes application of dCas9 to design diverse transcriptional regulatory circuits in E.coli. This includes several "classical" circuits such as a bistable switch, IFF loop and an oscillator. Results in terms of the performance of dCas9/sgRNA instead of natural and designed DNA binding proteins used previously are quite impressive, although it is not clear how some of those designs could function as they apparently violate some of the design principles. For publication in NatCommun this should be elucidated.

I have the following specific comments:

1. Repression by DCas9 downstream from the promotor has been shown before. However this depends strongly on the distance from the promotor, while the results in this MS show very similar repression by different sgRNAs. Sequences of all annotated DNA constructs should be provided in the supplement in order to enable others to check the design and replicate the study, for this segment but also for all presented results. It would also be interesting to identify the specific context of the *csy4* cleavage sequences. Are all cleavage sites really necessary ?
2. Demonstration of hysteresis is not sufficient to claim functional bistable switch. There are many potential reasons for the occurrence of hysteresis – from the reporter protein degradation rate, remaining inducer in the medium etc. Authors should demonstrate the time course of the two states and that the state remains stable even when the inducer has been removed.
3. Also as shown by theoretical analysis and experiments using monomeric DNA binding domains, nonlinear response, predominantly provided by cooperativity is required for the bistability. None of this seems to be present in the dCas9/sgRNA. If this system indeed exhibits a bistability authors should provide an explanation, although they argue that this is beyond the scope of this work.
4. Performance of the bandpass filter by an IFFL looks very nice. What is the advantage of a double stripe phenotype if they occur at exactly the same concentration range. Demonstration of a double stripe phenotype would have been much more exciting by a noncoincident inducer responsiveness or responsiveness to two different inducers.
5. As authors also comment synchronization of cells by CRISPRilator is completely unexpected as there do not seem to be any soluble mediator that would transfer the information between cells. In the absence of a plausible and verifiable mechanism we cannot exclude the possibility of some experimental setup bias, particularly since the oscillations do not seem to dampen as in almost all other oscillators and they start without adding an inducer. Controls should have been performed e.g. by omitting some components of the oscillator. Does the frequency depend on the flow rate as in case of soluble mediator driven quorum-based oscillations (Danino, Nature 2010)?

Reviewer #2:

Remarks to the Author:

This paper demonstrates that some of the standard example circuits in synthetic biology can be constructed in a very clean and straightforward way with CRISPRi components. CRISPRi is desirable compared to protein based regulation because it is much easier to program, or re-wire, circuits. Thus, using essentially the same types of parts, the authors build a not gate, a bistable switch, an IFFL, and a repressilator. They authors employ a number of good engineering techniques, such as putting the entire circuit on one plasmid and using orthogonal degradation tags, to get what look like very nice behaviors from circuits that are often quite difficult to tune. One of the main assumptions the authors test, and disprove, is that CRISPRi is might not exhibit sharp enough deactivation (sometimes called cooperatively) to make these circuits work.

It is been 19 years since the original bistable switch and repressilator were published. The paper shows we have simultaneously come a long way with building predictable circuits, while also showing that we have not come that far. We are still tuning circuits with three components without applying them to real world applications.

In terms of impact, this paper clearly shows that these circuits can be more easily engineered than they were 19 years ago. However, it does not provide much insight as to why other than relying on what seem like better informed, and more rational choices about some specifics and some unsupported (but believable) claims about their circuits incurring lower metabolic load. There is no mathematical modeling, for example, showing a robustness analysis of the old architecture versus the new (this is just an example of something that I would qualify as a deeper analysis).

Some specific comments:

Measuring hysteresis using cells pre-cultured in one of the inducers, is an interesting way of characterizing the switches. However, other papers presenting bistable switches typically should show a time series [inducer one, no inducer, inducer two, no inducer]. So the data here are hard to compare.

As for the oscillator: The say it rivals [40], in which the original repressilator was optimized to function much more robustly. It would be nice to see if the circuit in this paper works in liquid culture as in the corresponding one in [40] does. Otherwise it is hard to evaluate their claims.

Multistable and dynamic CRISPRi-based synthetic circuits

Javier Santos-Moreno, Eve Tasiudi, Jörg Stelling and Yolanda Schaerli

Point-by-point response to reviewers' comments

Reviewer #1 (Remarks to the Author):

Manuscript by Santos-Moreno and Schaerli describes application of dCas9 to design diverse transcriptional regulatory circuits in E.coli. This includes several “classical” circuits such as a bistable switch, IFF loop and an oscillator. Results in terms of the performance of dCas9/sgRNA instead of natural and designed DNA binding proteins used previously are quite impressive, although it is not clear how some of those designs could function as they apparently violate some of the design principles. For publication in NatCommun this should be elucidated.

We thank the reviewer for the constructive feedback. We addressed the comments and think that these changes improved the manuscript. We hope the reviewer agrees with us.

I have the following specific comments:

1. Repression by dCas9 downstream from the promotor has been shown before. However this depends strongly on the distance from the promotor, while the results in this MS show very similar repression by different sgRNAs.

To increase circuit modularity and predictability, all our sgRNA binding sites (“bs”) are at the same distance from the promoter. The manuscript now includes the following sentence: “All binding sites were placed at the same distance from the promoter to prevent differences in the repression derived from binding site position (Bikard et al 2013).”

Sequences of all annotated DNA constructs should be provided in the supplement in order to enable others to check the design and replicate the study, for this segment but also for all presented results.

All plasmids used in this study (and their complete sequences and maps) will be available through Addgene upon publication. We provide the sequences already for the reviewers.

It would also be interesting to identify the specific context of the csy4 cleavage sequences. Are all cleavage sites really necessary?

Given that each node in our circuits is a single transcriptional unit (TU) composed of multiple parts, a method is needed to ensure the independent functioning of the parts at the mRNA level, as opposed to “polycistronic” mRNA that may impact circuit modularity and predictability. We chose Csy4 cleavage (as opposed to e.g. ribozymes) due to the short (20bp) encoding space, also reducing the risk of recombination between multiple copies. The Csy4 cut sites have three functions: i) splitting two different sgRNAs encoded in the same TU (analogous to what occurs in the natural CRISPR array, where multiple crRNAs are transcribed together and further “individualized” by a Csy4 homologue), ii) splitting a sgRNA from an upstream transcribed

binding site, and iii) releasing the fluorescent reporter transcript (RBS + CDS) from upstream transcribed sequences (namely binding sites and sgRNAs). We think these mRNA splitting processes are required for reducing context-dependent effects and achieving modular and predictable circuit designs. However, we do not exclude that the circuits would still function even in the absence of some those cleavage sites. We now extend the description of the Csy4 cleavage site in the manuscript: “Csy4 is an RNase of the CRISPR system of *Pseudomonas aeruginosa* that processes the transcript of the CRISPR array to release the CRISPR-derived RNAs (crRNAs)³⁰. Here, we employed Csy4 processing to release parts that are transcribed together in the same mRNA molecule, but need to act independently once transcribed – e.g. sgRNAs, that need to bind dCas9 for function, and fluorescent reporter transcripts that have to be translated.”

2. Demonstration of hysteresis is not sufficient to claim functional bistable switch. There are many potential reasons for the occurrence of hysteresis – from the reporter protein degradation rate, remaining inducer in the medium etc. Authors should demonstrate the time course of the two states and that the state remains stable even when the inducer has been removed.

With the circuit topology we had in our initial manuscript, we could not perform the suggested experiment, as the inducers are required for the expression of the sgRNAs and the fluorescent reporters, and therefore the “no inducer” condition would result in no fluorescence rather than in the maintenance of the previous state. We had therefore called our network a “bistable circuit” and not a toggle switch. However, we now built a network where this is possible. To this end, we had to replace the inducible promoters with constitutive promoters (which do drive gene expression under the “no inducer” condition), and we used a double-inducible controller plasmid to toggle between the two states – a design that resembles more the original TS by Gardner and colleagues. We now call the network a toggle switch. In Fig. 2 we now show this new network and a time course experiment in which cells carrying the toggle switch (TS) circuit (and also two monostable controls, cL and cR) are subjected to an induction scheme that is typically used for TS characterization and demonstration of bistability: inducer 1 – no inducer – inducer 2 – no inducer.

The toggle switch section now reads as follows:

“We next sought to build a classical genetic circuit that has never been constructed before using CRISPRi: the toggle switch (TS)²⁶, where two nodes mutually repress each other to produce bistability, i.e. two mutually-exclusive stable states. We designed a TS circuit in which two nodes produce sgRNAs that repress each other (Fig. 2), resulting in two non-concurrent stable states: “HIGH”, in which the fluorescent reporter sfGFP-MarAn20 is produced at high levels, and “LOW”, in which sfGFP-MarAn20 is expressed at negligible levels. The two states can be toggled through a controller plasmid, in which the addition of Ara or AHL (N-(β -Ketocaproyl)-L-homoserine lactone) triggers the expression of sgRNA-1 or -4, thus favoring the LOW and the HIGH states, respectively (Fig. 2b). The TS was assessed by incubating cells in four successive media containing the first inducer (Ara or AHL), followed by no inducer, then the second inducer (AHL or Ara), and finally no inducer again (Fig. 2c and d). Most cells adopted the LOW state following an initial incubation with Ara, and kept this state even when Ara was removed; later, cells switched to the HIGH state when AHL was added, and kept the HIGH state after AHL removal (Fig. 2c left, and d). The reverse induction scheme (AHL-nothing-Ara-nothing) also displayed bistability and hysteresis (Fig. 2 right). Conversely, the control circuits lacking one of the repressions (Fig. 2a) failed to display any bistability: both cL (biased towards the HIGH state) and cR (biased towards the LOW state) showed a monostable behavior

and were unable to keep the LOW and the HIGH states in the absence of inducer, respectively (Fig. 2c). Together, these results show that CRISPRi can be applied to build multistable gene circuits.”

We have also modified accordingly the legend of Fig. 2 and the corresponding section of the methods.

3. Also as shown by theoretical analysis and experiments using monomeric DNA binding domains, nonlinear response, predominantly provided by cooperativity is required for the bistability. None of this seems to be present in the dCas9/sgRNA. If this system indeed exhibits a bistability authors should provide an explanation, although they argue that this is beyond the scope of this work.

We thank the reviewer for asking for a mechanistic explanation, which we now address by mathematical modeling / systems theory analysis for the toggle switch.

Briefly, theoretical studies such as Cherry & Adler, J. Theor. Biol 203: 177 (2000) prove that a two-gene circuit with mutual inhibition in the form of hyperbolic (Michaelis-Menten type) kinetics cannot generate bistability, whereas Hill kinetics with cooperativity can. The protein-based classical toggle switch was designed and implemented according to this principle to achieve the necessary non-linearity.

However, Cherry & Adler (2000) and subsequent studies showed that cooperativity is sufficient but not necessary for bistability in a two-gene circuit with mutual inhibition. Alternative mechanisms include function of the repressor as a dimer, non-additive effects of multiple binding sites on the DNA, and depletion of free repressor, for example, by specific or unspecific binding outside of the control region of the target gene. With dCas9 and a single engineered binding site in our constructs, the first two alternatives could not apply. However, the third alternative appeared plausible because of dCas9’s scanning of PAM sites – the effective affinity of binding to sites not matching the sgRNA sequence is comparatively low (considered here as unspecific binding), but such sites are abundant in the genome.

To evaluate this possibility, we used methods from chemical reaction network theory to analyze several mechanistic models (in the form of semi-diffusive chemical reaction networks) for the CRISPRi-based toggle switch. This analysis yielded the following key results:

1. For CRISPRi-based circuits with only specific binding to target genes indeed we did not find bistability for any (a priori unknown) parameter values in wide, biologically plausible parameter ranges.
2. Extensions of the network by unspecific binding to target genes, where specific and unspecific binding events are independent of each other (no cooperativity), are sufficient to generate bistability, even when unspecific binding does not inhibit target gene expression.
3. The above also holds when parameter values are constrained to experimentally determined ranges, wherever those were available in the literature.

Thus, our model-based analysis suggests that the interplay between specific and unspecific binding of dCas9/sgRNA complexes explains bistability of the CRISPRi-based toggle switch.

We have now added a corresponding figure (Fig. 3) and paragraph on “Origin of bistability” (inserted below) and provide all details on models and analysis in the methods, Supplementary Tables 2-3 and Supplementary Figs. 7-13.

“Because CRISPRi’s non-cooperative binding⁸ contrasts with the requirement for cooperativity in the protein-based TS²⁶, we aimed to explain how CRISPRi can generate multistability. Theory shows that cooperativity is sufficient, but not necessary for a two-gene circuit with mutual inhibition to be bistable; alternative mechanisms include a significant depletion of free repressor³⁴. We considered this a plausible mechanism for CRISPRi because of dCas9’s scanning of PAM sites – the effective affinity of binding to sites not matching the sgRNA sequence is comparatively low³⁵ (considered here as unspecific binding), but PAM sites are abundant in the genome. We therefore developed mechanistic mathematical models in the form of reaction networks with mass-action kinetics that comprise dCas9/sgRNA binding to DNA as well as production and degradation of all components (see Supplementary Fig. 7-12 for details). We then used methods from chemical reaction network theory^{36, 37} to evaluate the potential of a network structure for bistability, without prior knowledge on parameter values. Consistent with theory, we did not find bistability for networks with specific binding only. However, network extensions by unspecific binding to target genes, with independent (that is, non-cooperative) specific and unspecific binding events (Fig. 3a) could show bistability. This held even when unspecific binding did not inhibit gene expression and when we constrained parameters to experimentally determined ranges (Supplementary Table 3), as illustrated in Fig. 3b. Thus, our model-based analysis suggests that the interplay between specific and unspecific binding of dCas9/sgRNA complexes to DNA explains bistability in the CRISPRi TS.”

4. Performance of the bandpass filter by an IFFL looks very nice. What is the advantage of a double stripe phenotype if they occur at exactly the same concentration range. Demonstration of a double stripe phenotype would have been much more exciting by a noncoincident inducer responsiveness or responsiveness to two different inducers.

The double stripe results show that it is relatively straightforward to combine CRISPRi circuits that function orthogonally within the same cellular environment. Moreover, starting from a CRISPRi design one can easily create a new, orthogonal version of that design that features a very similar behavior to the original circuit, as expected. This allows for high predictability when constructing new CRISPRi circuits, and we think it is still worth highlighting. Nevertheless, we agree with the reviewer in the interest of combining two orthogonal CRISPRi circuit that display different behaviors. The manuscript now shows the combination of two CRISPRi circuits with different functions: a stripe-forming IFFL network and a double-inverter circuit (Fig. 4h).

The manuscript now includes the following paragraph: “We also assessed the combination of two CRISPRi circuits exhibiting different functions within the same cellular environment. We combined a stripe-forming IFFL with an orthogonal double-inverter circuit³³, a concatenation of two NOT gates that inverts twice the input signal to give an output that mimics the input. This design can be used when the output needs to resemble the input with some additional modification, e.g. a shifted position³³ or a time delay¹⁴. Bacteria carrying the two circuits perform both functions: an IFFL-generated stripe pattern and a double-inverter-driven behavior that mimics the Ara gradient (Fig. 4g and h). Thus, the two networks behaved as expected,

demonstrating the possibility to combine multiple CRISPRi circuits with distinct functions in the same bacteria.” Fig. 4 has been adapted accordingly.

5. As authors also comment synchronization of cells by CRISPRilator is completely unexpected as there do not seem to be any soluble mediator that would transfer the information between cells. In the absence of a plausible and verifiable mechanism we cannot exclude the possibility of some experimental setup bias, particularly since the oscillations do not seem to dampen as in almost all other oscillators and they start without adding an inducer. Controls should have been performed e.g. by omitting some components of the oscillator. Does the frequency depend on the flow rate as in case of soluble mediator driven quorum-based oscillations (Danino, Nature 2010)?

We acknowledge the reviewer for asking us to clarify regarding the synchrony of the oscillations. In fact, as the reviewer mentions, there is no active mechanism that would account for the observed synchronization. Instead, as in the optimized versions of the protein-based repressilator (Niederholtmeyer 2015 and Potvin-Trottier 2016), the CRISPRilator synchrony is likely coming from the robust inheritance of the period with minimal de-phasing due to stochastic molecular fluctuations. The microfluidic chambers were initially loaded with a low number of cells (2-3) that divided to fill the whole chamber, after which oscillations were determined for the whole population of bacteria (~110 cells) within the chambers. As “founder” cells divided, they kept the period of the oscillations very precisely and transmitted it to the daughter cells, resulting in a lineage of cells with synchronous oscillations. One of the lineages stochastically occupying most of the chamber, or all founder cells being in the same phase of the oscillation, both result in microfluidic chambers with hundreds of cells in synchrony. In Danino et al. 2010, the period changes depending on the flow rate because the oscillator is activated by a soluble mediator (AHL) that is swept away by the fluid flow, which is not the case for the CRISPRilator.

The manuscript has been adapted to showcase the fact that it is the period robustness of the CRISPRilator that results in synchronous oscillations.: “Few (2-3) bacteria carrying the CRISPRilator were loaded per microfluidic chamber (Supplementary Fig. 5) and grown there continuously for 3 days (over 100 generations) and imaged every 10 min. The CRISPRilator generated robust oscillations between three states (“red”, “green” and “blue”) with a period of 10-12 h (14-17 generations, Fig. 5c and d), while a control with an “open-ring” topology does not show any oscillations (Supplementary Fig. 6). After the chambers were filled with cells (~110 cells/chamber), we observed synchronous long-term oscillations, even in the absence of any active cell communication or synchronization mechanism (Supplementary Movie 1). This is reminiscent of improved versions of the protein-based repressilator^{40,41}, where synchronous oscillations were observed due to lower levels of molecular fluctuations and consequently reduced phase drift and increased inheritance of the period compared to the original repressilator²⁷.”

The supporting information now also includes a control for the CRISPRilator in which one of the sgRNAs (sgRNA-4) was replaced by a different sgRNA (sgRNA-3) lacking any binding site within the circuit, which results in an “open-ring” architecture as opposed to the “closed-ring” repression topology of the CRISPRilator (Supplementary Fig. 6). This control shows no sign of an oscillatory behavior (Supplementary Fig. 6).

Reviewer #2 (Remarks to the Author):

This paper demonstrates that some of the standard example circuits in synthetic biology can be constructed in a very clean and straightforward way with CRISPRi components. CRISPRi is desirable compared to protein based regulation because it is much easier to program, or re-wire, circuits. Thus, using essentially the same types of parts, the authors build a not gate, a bistable switch, an IFFL, and a repressilator. They authors employ a number of good engineering techniques, such as putting the entire circuit on one plasmid and using orthogonal degradation tags, to get what look like very nice behaviors from circuits that are often quite difficult to tune. One of the main assumptions the authors test, and disprove, is that CRISPRi is might not exhibit sharp enough deactivation (sometimes called cooperatively) to make these circuits work.

It is been 19 years since the original bistable switch and repressilator were published. The paper shows we have simultaneously come a long way with building predictable circuits, while also showing that we have not come that far. We are still tuning circuits with three components without applying them to real world applications.

We thank the reviewer for appreciating that we are able to build predictable circuits using CRISPRi. The reviewer is correct, this manuscript does not show any real-world applications. The primary aim of this publication is to demonstrate that CRISPRi can be used to build multistable and dynamic circuits – contrary to the common belief. However, the favorable properties of CRISPRi combined with our circuit design (programmability, orthogonality, low burden, short encoding sequences, high modularity) make these circuits highly interesting for applications. It will also enable the construction of further synthetic CRISPRi circuits suitable for applications. Indeed, since the posting of the pre-print we have already been contacted by several potential collaborators that would like to use our CRISPRi circuits for their applications.

In terms of impact, this paper clearly shows that these circuits can be more easily engineered than they were 19 years ago. However, it does not provide much insight as to why other than relying on what seem like better informed, and more rational choices about some specifics and some unsupported (but believable) claims about their circuits incurring lower metabolic load.

We now provide evidence supporting the claim that the metabolic load of CRISPRi circuits is low. In Supplementary Fig. 3 we show that bacteria carrying two stripe networks (Fig. 4e and f) have similar growth rates as cells carrying only one of the stripe circuits or no stripe circuit at all. We have previously built protein-based stripe-forming networks (Schaerli et al., Nat. Commun. 2014). From this, we know that even introducing an additional protein (e.g. a reporter protein) causes severe metabolic load. In our hands it was not possible to have a protein-based stripe-forming network with three fluorescent reporters, let alone two stripe-forming networks in the same cell. Using CRISPRi thus opens completely new possibilities in increasing the complexity of synthetic circuits.

There is no mathematical modeling, for example, showing a robustness analysis of the old architecture versus the new (this is just an example of something that I would qualify as a deeper analysis).

We now include a deeper analysis based on mechanistic mathematical modeling and application of chemical reaction network theory. It highlights a potential source of the non-linearity in CRISPRi that enables us to build multistable circuits (a combination of specific

binding of dCas9/sgRNAs to their target sequences and ‘unspecific’ scanning of PAM sequences in the genome). Please see our answer to reviewer #1, point 3 for details.

A formal analysis of robustness, comparing old and new architecture, as suggested by the reviewer, could certainly be interesting. However, we argue that a systematic analysis would have to include at least detailed molecular mechanisms and stochastic effects, as illustrated by the comparison of the original repressilator and the improved repressilator by Potvin- Trottier (2016) - they differ in apparently minor, but decisive molecular details. Therefore, we consider this beyond the scope of the present study.

In addition, we would like to point out that we do not claim that the CRISPRi circuits have higher robustness than protein-based circuits. We think the fact alone that CRISPRi can be used for generating dynamic and multistable behaviors is surprising and worth reporting. Combined with the benefits inherent to CRISPRi circuits (programmability, orthogonality, low burden, short encoding sequences) this enables the construction of synthetic circuits that would be impossible or at least hard to build with protein repressors. We will report such circuits in future publications.

Some specific comments:

Measuring hysteresis using cells pre-cultured in one of the inducers, is an interesting way of characterizing the switches. However, other papers presenting bistable switches typically should show a time series [inducer one, no inducer, inducer two, no inducer]. So the data here are hard to compare.

This comment was also made by reviewer #1. Please see our answer to point 2 of reviewer #1.

As for the oscillator: The say it rivals [40], in which the original repressilator was optimized to function much more robustly. It would be nice to see if the circuit in this paper works in liquid culture as in the corresponding one in [40] does. Otherwise it is hard to evaluate their claims.

Please see also the response to point 5 of reviewer #1. The oscillations in the Potvin-Trottier repressilator (Potvin-Trottier et al. 2016) can be paused by adding IPTG to the medium. Once IPTG is removed, all cells will start their oscillations in phase. Combined with a robust period this enables to detect the oscillations in liquid culture. The CRISPRiator we report in the manuscript does not allow us to do the same and pause the oscillations by a small molecule in the medium. However, we built a version where one promoter is Ara-inducible. Preliminary data indicate that we can observe the oscillations in liquid culture with this circuit.

Nevertheless, we now have rewritten this paragraph to remove the statement that it rivals the Potvin-Trottier repressilator:

“After the chambers were filled with cells (~110 cells/chamber), we observed synchronous long-term oscillations, even in the absence of any active cell communication or synchronization mechanism (Supplementary Movie 1). This is reminiscent of improved versions of the protein-based repressilator^{46, 47}, where synchronous oscillations were observed due to lower levels of molecular fluctuations and consequently reduced phase drift and increased inheritance of the period compared to the original repressilator²⁸.”

Reviewers' Comments:

Reviewer #1:

Remarks to the Author:

The revised manuscript is indeed improved in many ways and I think the authors probably appreciate it as well. The explanation for the bistability based on nonspecific binding seems plausible and corroborated by a model. nevertheless the title of the Fig3 might more appropriately be: Proposed origin of bistability...

With respect to the sequences I would nevertheless suggest to include the sequences of all constructs into the Supplementary material rather than at the Addgene submissions.

The only nonresolved issue for this reviewer are the sustained synchronous oscillations. The explanation that this is due to the robust inheritance across several cell divisions seems a bit far fetched. Without imposing excessive additional experiments to the authors it should be relatively easy to test this hypothesis by a simple experiment: chamber should be seeded by a mix of cells from two transfections performed with a certain (say 5 h) delay. If their hypothesis is correct they should be able to observe a superposition of two oscillations (with amplitude proportional to each cell type fraction), which would make the paper even more interesting.

Reviewer #2:

Remarks to the Author:

This new version of the paper improves upon the previous version in many ways, especially with the inclusion of a toggle switch and more standard experimental demonstration that it works. Other changes made in response to the reviews seem solid to me. I apologize for not being able to write a more thorough review at this time.

Multistable and dynamic CRISPRi-based synthetic circuits

Javier Santos-Moreno, Eve Tasiudi, Jörg Stelling and Yolanda Schaerli

Point-by-point response to reviewers' comments

Reviewer #1 (Remarks to the Author):

The revised manuscript is indeed improved in many ways and I think the authors probably appreciate it as well. The explanation for the bistability based on nonspecific binding seems plausible and corroborated by a model. Nevertheless the title of the Fig3 might more appropriately be: Proposed origin of bistability...

We agree with the reviewer on the improvement of the manuscript and acknowledge her/him for the valuable feedback. The title of Fig. 3 legend is now “Proposed origin of bistability in the CRISPRi TS” as suggested.

With respect to the sequences I would nevertheless suggest to include the sequences of all constructs into the Supplementary material rather than at the Addgene submissions.

Full plasmid sequences are now provided as a separate Supplementary Data 1. We do not like to include them in the normal Supplementary information in order to avoid that somebody prints by mistake the 165 pages out.

The only nonresolved issue for this reviewer are the sustained synchronous oscillations. The explanation that this is due to the robust inheritance across several cell divisions seems a bit far fetched. Without imposing excessive additional experiments to the authors it should be relatively easy to test this hypothesis by a simple experiment: chamber should be seeded by a mix of cells from two transfections performed with a certain (say 5 h) delay. If their hypothesis is correct they should be able to observe a superposition of two oscillations (with amplitude proportional to each cell type fraction), which would make the paper even more interesting.

Unfortunately, due to the COVID-19 lockdown we are currently unable to perform this experiment in the expected time frame. However, we do not think that this explanation is far-fetched. Other implementations of the repressilator previously demonstrated robust inheritance of the period across cell divisions (Niederholtmeyer, H. *et al.* eLife 4, e09771, 2015 and Potvin-Trottier, *et al.* Nature 538, 514-517, 2016).

Moreover, when we seed the microfluidic chambers with few cells (1-3), after an initial period, we observe easily oscillations synchronized across the population in the microfluidic chamber as shown in the manuscript. However, when we seed the chambers with more cells (20-30) there is a higher heterogeneity in the number of initial oscillation states. With more initial cells, it is less likely that an individual lineage takes over the whole chamber and consequently, the oscillations are less synchronized across the chamber. However, cells from the same lineage clearly inherit the period of their mother cells (see provided movie for review).

On the other hand, we think that the proposed experiment would not clarify this issue, since not all the cells resulting from a single transformation share the same oscillation state, and therefore two sequentially transformed populations might well be heterogeneous and indistinguishable from each other.

We added now the following sentence to tune down the claims related to the synchronous oscillations: “We hypothesize that the observed synchrony stems from the robust inheritance of the oscillatory state across cell divisions from the few cells initially seeded in the microfluidic chamber.”

Reviewer #2 (Remarks to the Author):

This new version of the paper improves upon the previous version in many ways, especially with the inclusion of a toggle switch and more standard experimental demonstration that it works. Other changes made in response to the reviews seem solid to me. I apologize for not being able to write a more thorough review at this time.

We acknowledge the reviewer for the feedback and agree on the manuscript's improvement.